



# Impact of agricultural interventions on ammonia emissions and on PM$_{2.5}$ concentrations in the UK: a local and regional modelling study

Matthieu Pommier[1], Robert Benney[2], Jamie Bost[1], Becky Jenkins[3], Joe Richardson[4], Liam Rock[4], Olivia Blythe[4], Oliver Marshall[2], Alexandra Spence[2]

[1]Ricardo Energy & Environment, 18 Blythswood Square, Glasgow G2 4BG, UK.
[2]Ricardo Energy & Environment, Bright Building, Manchester Science Park, Pencroft Way, Manchester M15 6GZ, UK.
[3]Ricardo Energy & Environment, The Gemini Building, Fermi Avenue, Harwell, Didcot OX11 0QR, UK.
[4]Ricardo Energy & Environment, 30 Eastbourne Terrace, London W2 6LA, UK.

*Correspondence to*: Matthieu Pommier (matthieu.pommier@ricardo.com)

**Abstract.** The contribution of agricultural emissions of fine particulate matter (PM$_{2.5}$) poses significant health and environmental challenges, particularly in the UK where intensive farming activities contribute to elevated pollutant levels. This contribution includes direct emissions and PM$_{2.5}$ formed through chemical reactions from precursors such as ammonia (NH$_3$). The study aims to analyse the impact of series of mitigation measures through emission scenarios (low, medium, high uptake) on dairy, pig and poultry sectors in 2030 and mainly focusing on NH$_3$ emissions. Under the high uptake scenario, NH$_3$ emissions could decrease by up to 13% nationally, with reductions reaching as high as 20% in certain regions. The Community Multiscale Air Quality (CMAQ) and the Atmospheric Dispersion Modelling System (ADMS) models were used. CMAQ allows to understand the contribution made by agricultural NH$_3$ to secondary PM$_{2.5}$ at a regional scale, while ADMS is used to better understand near-field dispersion and dilution of primary pollutants. Despite the impact of the changes in emissions due to the mitigation measures compared to the future baseline scenario, changes are not reflected on regional scale PM$_{2.5}$ concentrations since the maximum modelled decrease was around 1-1.5%. This finding is explained by an NH$_3$-rich atmosphere reducing the impact of these reductions in NH$_3$ emissions on mitigating PM$_{2.5}$ concentrations. Results from ADMS show that the NH$_3$ and PM$_{2.5}$ concentrations are quickly dispersed near the farms, highlighting the usefulness of local modelling in addressing impact studies on PM$_{2.5}$ formation near these sources. Indeed, for the five studied livestock farms, it has been found that 50% of maximum NH$_3$ and PM$_{2.5}$ concentrations are located within a distance between 100 and 400m and up to 90% of concentrations have decreased within 700m. The study also demonstrates the complementary use of local and regional modelling in understanding PM$_{2.5}$ dispersion near agricultural areas. The comparison with ground-based measurements might suggest a non-representation of atmospheric processes in the PM$_{2.5}$ formation by CMAQ (with an underestimation of PM$_{2.5}$ concentrations by approximately 50%). It underscores the need for integrated modelling approaches to guide mitigation strategies for both primary and secondary PM$_{2.5}$, as well as to improve understanding of the chemical atmospheric processes involved in the secondary inorganic aerosols.



# 1   Introduction

Air pollution from $PM_{2.5}$ (fine particulate matter with a mass median aerodynamic diameter <2.5 µm) has been estimated to cause millions of premature deaths annually in recent years (Burnett et al., 2018; Kiesewetter et al., 2015; Lelieveld et al., 2015). $PM_{2.5}$ poses significant environmental and public health problems due to its ability to penetrate deep into the respiratory system, causing various health issues, including respiratory and cardiovascular diseases (Pope and Dockery, 2006). Therefore, mitigating this $PM_{2.5}$ pollution is a high priority for environmental protection in many areas such as the European Union (EU) and in the United Kingdom (UK).

Among the various components contributing to $PM_{2.5}$ concentrations, ammonia ($NH_3$) has an important role in secondary particulate formation. In the atmosphere, $NH_3$ reacts with acidic compounds such as sulfuric acid ($H_2SO_4$) and nitric acid ($HNO_3$), forming ammonium sulphate (($NH_4)_2SO_4$) and ammonium nitrate ($NH_4NO_3$), which are significant constituents of $PM_{2.5}$ (Seinfeld and Pandis, 2016; Wyer et al., 2022).

Resulting of its varied agricultural practices, transport-related emissions, and industrial activities, the UK presents a significant case for examining the influence of ammonia ($NH_3$) on $PM_{2.5}$ levels. $NH_3$ emissions in the UK primarily originate from agricultural sources, particularly livestock waste and the application of fertilizers (Misselbrook, et al., 2023). Indeed, the most recent figure from the UK National Atmospheric Emissions Inventory (NAEI) shows that agriculture accounted for 87% of total ammonia emissions in 2021 (NAEI, 2024). These emissions have been shown to vary seasonally and spatially, influencing the formation and distribution of airborne $PM_{2.5}$ concentrations (e.g. Wyer et al. 2022). Various mitigation measures (i.e. farm practices) have been developed to mitigate emissions of $NH_3$, such as covering slurry stores, or using automatic scrapers in housing, however, reducing air pollution from agriculture remains challenging (Jenkins and Wiltshire, 2024).

Previous studies have highlighted the importance of understanding the interaction between $NH_3$ and $PM_{2.5}$ to inform regulatory measures and mitigate adverse health effects. For instance, the work by Vieno et al. (2014) demonstrated that reductions in $NH_3$ emissions could lead to significant decreases in $PM_{2.5}$ levels, especially in areas with large nitrogen oxides ($NO_x$) concentrations, suggesting that targeted strategies in $NH_3$ emission control could be effective in improving air quality. Results confirmed by the study of Ge et al. (2023) showing $NH_3$ reductions are more effective for regions or countries with better air quality, such as in the UK (compared to Asia, for example) to mitigate $PM_{2.5}$ concentrations. The impact of $NH_3$ emissions reduction is significantly more efficient with large emission reduction measures (Bessagnet et al., 2014) and abating $NH_3$ emissions can even be more cost-effective than $NO_x$ for mitigating $PM_{2.5}$ air pollution (Gu et al., 2021). Conversely, other work such as Ge et al., (2022) and Pay et al. (2012), suggested $NH_3$ emissions reduction may only lead to minor improvements in airborne $PM_{2.5}$ concentrations, especially in the UK since the UK is characterized by a $NH_3$-rich atmosphere. A study in the United States also showed controlling $NH_3$ became significantly less effective for mitigating $PM_{2.5}$ in rural areas (Pan et al., 2024).

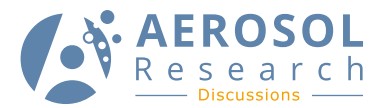

Due to the complexity of atmospheric chemistry, numerical air quality models such as Chemistry Transport Models (CTMs) are commonly used to simulate these processes and assess the effectiveness of potential emission control strategies. CTM such as the Community Multiscale Air Quality (CMAQ) model (Appel et al., 2021), developed and distributed by the US Environmental Protection Agency (EPA) is a cutting-edge numerical air quality model that comprehensively represents the emission, formation, destruction, transport, and deposition of numerous air pollutants, including $PM_{2.5}$ and its precursors. CTMs such as CMAQ are designed to calculate background concentrations, i.e. air pollutants' concentrations at a km scale spatial resolution (De Visscher, 2014).

Local dispersion models like Atmospheric Dispersion Modelling System (ADMS) (Carruthers et al., 1994) can be utilized to provide detailed simulations of pollutant dispersion at a finer scale such as 1m. ADMS is particularly effective for assessing the impact of emissions from specific sources and understanding local air quality variations (Zhong et al., 2023). The combination of local dispersion models such as ADMS with CTMs allows a more comprehensive understanding of both regional and local air quality dynamics. Indeed, local modelling studies have shown their accuracy in determining the dispersion of pollution (Hood et al., 2018; Porwisiak et al., 2024; Zhong et al., 2023). ADMS is by default a steady state (non-reactive) Gaussian plume model that predicts pollutant concentrations based on the assumption that both the vertical and horizontal dispersion of the continuous plume is represented by normal distribution around the plume centreline. However, due to the steady state assumption, short range estimates within 10km are recommended (Environmental Protection Ireland Agency, 2020).

The aim of the study was to understand the impact of mitigation measures relating to livestock housing and the storage and spreading of manures and slurries on $PM_{2.5}$ concentrations and was part of an interdisciplinary project named AIM-Health (Cowie et al., 2025). A companion study has already presented the impact of these policies on $NH_3$ concentrations and nitrogen deposition at a regional scale (Pommier et al., 2025). This study primarily focussed on measures to reduce emissions from housed dairy, pigs and poultry, while emissions from other sources such as manufactured fertilisers were not within its scope. Three intervention scenarios were developed to model the impact on $PM_{2.5}$ concentrations nationally based on differing uptake levels of the mitigation measures across the UK, ranging from low, medium and high. Additionally, local modelling was done to show how primary emissions of $NH_3$ and $PM_{2.5}$ disperse within the local vicinity (10km) of farms included in this study.

Section 2 of this paper describes the methodology used for the scenario development and the air quality modelling (regional and local). The analysis on the modelled $PM_{2.5}$ concentrations is presented in Section 3. Section 4 discusses the results and Section 5 gives the conclusions.

## 2    Method

A series of mitigation measures related to livestock diet, livestock housing and improved storage and spreading of manures and slurries were modelled to understand the impact on emissions from housed dairy, pigs and poultry across the UK. The





mitigation measures were modelled through scenarios which represented various levels of uptake (low-high) on these farms
across the UK in 2030.

To undertake the study, the CMAQ model, has been used for the regional modelling. CMAQ is a 3D Eulerian model, incorporating the effects of meteorology, emissions, land use, chemistry and aerosol processes on modelled air pollution. It has been developed to represent the emission, transport, formation, destruction, and deposition of many air pollutants, including nitrogen dioxide ($NO_2$), ozone ($O_3$) and $PM_{2.5}$. The version used in this study is 5.4 (US EPA Office of Research

and Development; https://zenodo.org/records/7218076, 2022a). This chemical-transport model requires input from a weather model, emissions and the background atmospheric composition. For our work, the CMAQ model has been driven by meteorological fields from the Weather Research and Forecasting (WRF) model version 4.5 (NCAR, 2022).

For the local modelling, ADMS version 6 (CERC, 2024) has been used. ADMS is steady-state Gaussian air dispersion model that incorporates air dispersion based on planetary boundary layer turbulence structure and scaling concepts, including

treatment of both surface and elevated sources, and both simple and complex terrain. This model allows calculation of concentrations of atmospheric pollutants emitted both continuously from point, line, volume and area sources, or intermittently.

## 2.1    Scenario development

The list of 19 mitigation measures were identified by European Commission's Best Available Techniques (BAT) reference

document for the intensive rearing of poultry or pigs (European Commission. Joint Research Centre., 2017) and Defra's Code of Good Agricultural Practice (COGAP) for Reducing Ammonia Emissions (DEFRA, 2024b). The year 2030 was chosen due to being 10-years in the future from the start of the research study, therefore establishing a realistic timeline for practical implementation of new activities on farms. These measures mainly focus on controlling $NH_3$ emissions and not on mitigating the primary $PM_{2.5}$ emissions from farming activities.

Three scenarios have been considered: low, medium and high uptake and compared to a baseline in 2030 and defined in the rest of the document as low2030, medium2030, high2030 and base2030, respectively. The uptake scenarios were developed through stakeholder engagement with farmers and stakeholders (i.e. farm advisers, academics and farmer representatives). Each scenario includes all 19 mitigation measures, however with varying percentages of uptake, a table presenting levels of uptake is presented in Appendix A and a table with descriptions of the mitigation measures is in Appendix B. The uptake

rates were unique to each mitigation measure in each sector and were reflective of feedback received through engagement activities. The engagement activities included an online survey, focus groups and one-to-one interviews with participants from the dairy, pig and poultry sectors and those in other sectors which utilise manure or slurry. A total of 161 people took part in the activities. Full results and methodology are detailed in Jenkins and Wiltshire (2024).

Discussions in these activities were centred around understanding the current level of uptake and the benefits and barriers

associated with the mitigation measures to determine a potential future uptake. If a mitigation measure was received positively, it was estimated to have a higher uptake compared to measures that were received negatively by participants. This



was determined in the final level of uptake for each scenario. The future uptake did not take account of any potential changes to legislation that may have an impact as this information is not known, additionally there were no different uptakes for each part of the UK due to a lack of data.

To determine the emission reduction associated with each mitigation measure, the Scenario Modelling Tool (SMT) was used (Ricardo EE, 2021). The SMT is a model for the management and analysis of complex scenarios of mitigation of air quality and greenhouse gas emissions from agricultural sources in the UK. In this work, the model implements a mass flow model to track pollutant transfer between each of the locations on a farm, to correctly reflect the cascade of mitigation effects along the manure management chain.

The SMT calculates the effect on emissions of each scenario by adding measures with emission reduction values and uptake rates. It allows designing mitigation measures using the effect on emissions (as a percentage reduction), cost, and targeting (the point in the agricultural system/manure management chain at which the effect on emissions is felt). Uptake rates are used in the SMT, allowing for the uptake of each measure to be reflected as a percentage of a cohort of farms (e.g., fixed slurry cover can be applied to 15% of dairy farms). It is worth noting that the cost impact of the measures is not discussed in

this study.

There are different ways that the various types of measures are calculated within the SMT. In this study, 'Emission' and 'Reduction' measures were used. 'Emission' measures directly reduce the pollutant emission factor at a location on a farm. This type of measure represents changes in practice or technical solutions and is not typically used where a measure represents a change in the overall management system. 'Reduction' measures reduce the quantity of a source of emissions

(e.g. the number of animals in housing or the quantity of excreta in housing). This reduction is reflected in emissions occurring at all associated locations. In this study, the only 'Reduction' measures used related to extended grazing on dairy farms and low protein diets in dairy, pig, and poultry farms. For the low protein diet measures the quantity of excreta was reduced, while for the extended grazing the quantity of managed solid and liquid manure was reduced. All other measures were implemented as 'Emission' measures; directly reducing the emission factors at relevant locations.

The SMT comes with a default library of mitigation measures and associated emission reduction factors. These emission reduction factors have been calculated based on empirical evidence and published scientific literature; primarily UK based, and with reference to relevant international studies and the UNECE Task Force for Reactive Nitrogen Ammonia Abatement Guidance Document (Bittman et al., 2014). The mitigation impact of these measures from the SMT is verified for accuracy by comparison with data from the Agricultural Ammonia and Greenhouse Gas Inventory (AAGHGI) (Misselbrook, et al.,

160    2023).

Eleven measures that were included in the modelling in this project were not included in the pre-defined measure library. This uses COGAP, BAT and expert knowledge to determine how to reflect these measures in the SMT (including what stage(s) in the agricultural system the measure is relevant to and if it is an 'Emission' or a 'Reduction' measure), as well as the emission reduction potential. This information was added to the SMT using the 'Measure' function as outlined above.



The calculation of measure effect takes account of measure interactions, including the order of implementation and exclusivity, and employ the principal of maximum overlap of uptake and a multiplicative effects model, in line with similar, earlier models such as National Ammonia Reduction and Strategies Evaluation System (NARSES) (Webb et al., 2006; Webb and Misselbrook, 2004). Baseline emission data comes from the AAGHGI (Misselbrook, et al., 2023). The data set for the year 2019 was used as baseline as it was the most recent submission at the time of running the scenarios.

## 2.2 Regional modelling: CMAQ

### 2.2.1 Model set-up

The CMAQ model, calculating the pollutants' concentrations and depositions, was setup using the same vertical and horizontal grid structure as for WRF, modelling the meteorology. Atmospheric chemistry was simulated using the carbon bond mechanism (CB06r5) (Luecken et al., 2019) combined with the aerosol mechanism using the 7th generation aerosol module (AERO7) (Pye et al., 2017). The configurations of the WRF and CMAQ models are given in Table 1.

**Table 1: Summary of WRF and CMAQ modelling settings**

| WRF configuration – version 4.5 | Scheme |
|---|---|
| Longwave radiation | Rapid Radiation Transfer Model Global (Iacono et al., 2008) |
| Shortwave radiation | Dudhia (Dudhia, 1989) |
| Planetary boundary layer | ACM2 (Pleim, 2007) |
| surface layer | Pleim (Pleim, 2006) |
| Land-Surface | Rapid Update Cycle (RUC) (Smirnova et al., 2016) |
| Cumulus | Kain-Fritsch (Kain, 2004) |
| Land use classification | Noah-modified 21-category IGBP-MODIS (Friedl et al., 2002) |
| CMAQ configuration - version 5.4 | Scheme |
| chemistry | Cb6r5 (Luecken et al., 2019) |
| aerosol | Aero7 (Pye et al., 2017) |
| aerosol deposition parameterization | M3Dry (Hogrefe et al., 2023) |

A nested modelling approach has been employed, dividing the broader geographic area into smaller domains to enhance spatial resolution. This hierarchical structure enables more accurate representation of variations in emissions and

meteorological conditions. The outer domain, covering Europe, uses a horizontal resolution of 50 km (EU50), while the inner domain focuses on the UK with a finer resolution of 10 km (UK10), as illustrated in Figure 1.





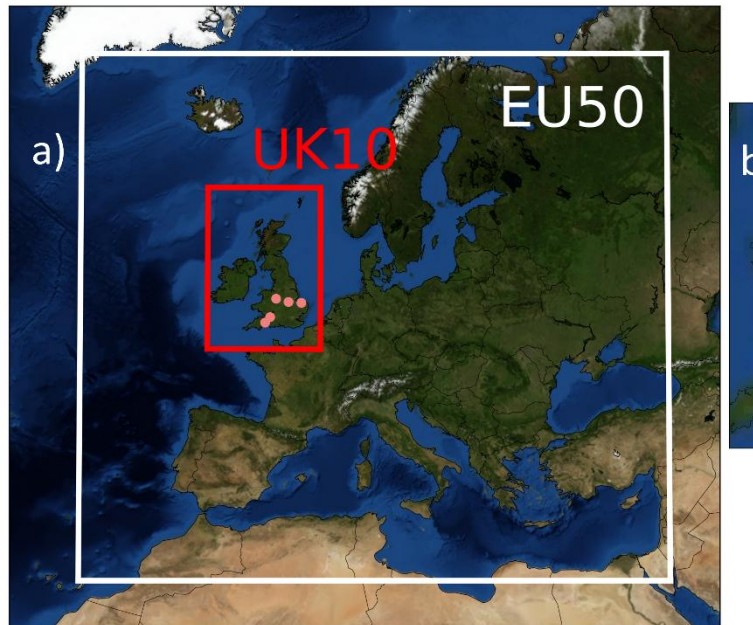
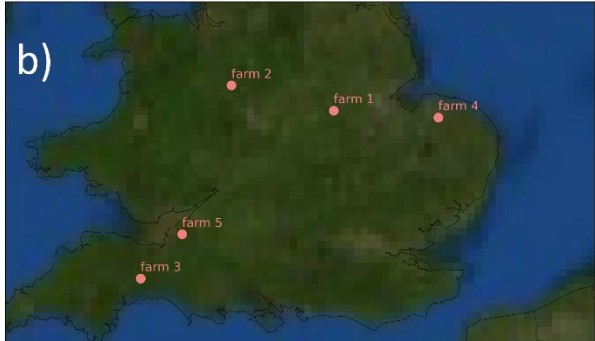

**Figure 1: a) Regional nested modelling domains and location of the studied farms. The white box corresponds to the European domain at 50 km × 50 km horizontal resolution (EU50) and the red box to the UK domain at 10 km × 10 km horizontal resolution (UK10). Each farm is shown with a pink coral circle. b) Zoom on the location of each studied farm with their corresponding id. The details on the farms are provided in Table 2.**

The selected meteorological year used in the air quality simulations was 2019. The year 2019 has been chosen as the reference year since it was defined as a typical meteorological year in the UK (see Pommier et al. 2025 and references within) and 2019 was also the most recent UK emissions year at the beginning of the project. This historical 2019 simulation has been used for model performance evaluation prior the analysis of the future predictions with the scenarios. The future scenarios solely focused on change in emissions and no climate projection has been undertaken.

The regional simulation started with a spin-up period of 2 weeks. The simulation setup follows a 'forecast-cycling' approach, where the output fields from each run were used to initialize the simulation for the following day. This process has been applied continuously throughout the entire year of 2019 for both the EU50 and UK10 domains. The initial and boundary conditions for the outermost domain (EU50) were created using hemispheric CMAQ outputs for the year 2016 provided by the US EPA (US EPA Office Of Research And Development, 2022b). Subsequently, the CMAQ concentrations computed within the EU50 domain were used as boundary conditions for the nested UK10 domain.

### 2.2.2 Emissions

The anthropogenic emissions data from the European Monitoring and Evaluation Programme (EMEP) (CEIP, 2022) were post-processed into 50 × 50 km to populate our EU50 domain in CMAQ. The UK anthropogenic emissions, including from agriculture, were based on the gridded emissions from the UK National Atmospheric Emission Inventory (NAEI) for 2019



(Churchill et al., 2021). The NAEI provides gridded emissions data at a 1 km × 1 km resolution, which was post-processed to match the 10 km × 10 km resolution of the UK10 domain. Additionally, the 2019 large point source emission inventory was used to vertically distribute emissions within the CMAQ grid.

The baseline 2030 future scenario for the EU50 domain was based on the EMEP gridded emissions for 2019 and scaled with the factors provided by the GAINS ECLIPSE (Greenhouse Gas and Air Pollution INteractions and Synergies - Evaluating the Climate and Air Quality Impacts of Short-Lived Pollutants) V6b Baseline CLE scenario (IIASA, 2019).

With the exception of the UK base2030 scenario, all UK scenarios incorporate the same set of measures. The increasing adoption of these measures across the low2030, medium2030, and high2030 scenarios reflects progressively higher ambition

in reducing air pollutant emissions as described in Section 2.1.

Figure 2 shows the total UK anthropogenic emissions as used in CMAQ and highlights the main changes in these emissions for the different scenarios. Since the mitigation measures mainly tackle the $NH_3$ emissions, this explains the large decrease calculated for this pollutant. As explained in Pommier et al. (2025), the reduction in $NH_3$ emissions could reach up to 20%, 22%, and 24% in certain regions under the low2030, medium2030, and high2030 mitigation scenarios, respectively.

A constant decrease in carbon monoxide (CO) is predicted across all scenarios. Unlike other pollutants, this trend is influenced not only by the selected mitigation measures but also by the scope of the SMT model, which does not fully capture all future CO emission sources. Slightly larger reductions in emissions are calculated for the high2030 scenario for volatile organic compounds (VOCs) and the coarse PM ($PM_{10}$, PM with an aerodynamic diameter lower than 10 μm), while the changes in $NO_x$ and $PM_{2.5}$ remain limited, and null for sulphur dioxide ($SO_2$).

CMAQ also calculates biogenic emissions with an online module incorporated in the model. This uses the Model of Emissions of Gases and Aerosols from Nature (MEGAN) (version 3.2) (Guenther et al., 2020). CMAQ also calculates windblow dust (Foroutan et al., 2017) and sea spray emissions (Gantt et al., 2015; Kelly et al., 2010) with online modules. These emissions are identical in all scenarios.



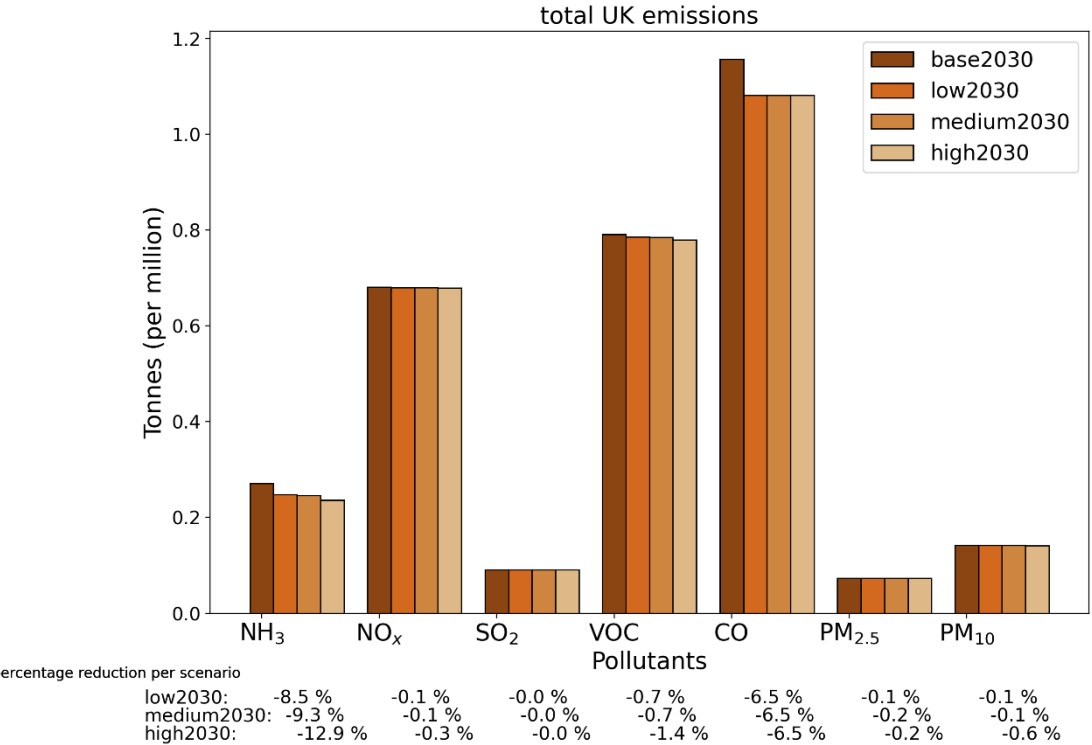

**Figure 2: Total UK anthropogenic emissions in tonnes for the different scenarios used by CMAQ for NH₃, NOₓ, SO₂, VOC, CO,**
**PM₂.₅, and PM₁₀. The relative difference for the low2030, medium2030 and high2030 scenarios compared to the base2030 are given below each corresponding bar.**

### 2.3 Local dispersion modelling: ADMS

#### 2.3.1 Model setup

For the local modelling, meteorological datasets were procured from National Oceanic and Atmospheric Administration
(NOAA) weather stations ranging from 6km to 25km from farms in this study, where data capture was poor filling was undertaken to ensure data capture is higher than 85% for all parameters including wind speed, wind direction, cloud cover, temperature and precipitation. 2019 was selected as this year is consistent with the existing baseline year of the regional model.

Each farm had a 15km-by-15km points grid centred at the farm with a 100m resolution. This was overlaid with the
CORINE Land Cover 2018 100m data (European Environment Agency, 2019) to extract map codes for each grid point. The land use classifications were associated with a surface roughness ranging between 0.04025 (water) and 1.3 (urban areas) in Aermet, the meteorological pre-processor for Aermod (Support Center for Regulatory Atmospheric Modeling, 2017 Appendix W Final Rule). NH₃ deposition was considered by using deposition velocities that vary depending on the surface. The deposition velocity values used for NH₃ vary between 0.02 m/s for lower plants (lowland shrubs, grassland) and 0.03



m/s for higher plants (woodlands) (Natural Resources Wales, 2021). Plume depletion was turned on in ADMS, this means that atmospheric concentrations of $NH_3$ and $PM_{2.5}$ decrease due to dry and wet deposition.

The requirement for complex terrain was established using the Environment Agency's 1m Lidar data (DEFRA, 2023) to see if it met Defra's Local Air Quality Management modelling requirement (> 1:10) (DEFRA, 2022) for any of the farms. None of the farms displayed a terrain of 1:10 or above and so complex terrain was omitted from the model.

ADMS can include buildings to simulate the impact of building downwash for point sources only, air recirculation leeward (downwind) of the building. Buildings within a distance three times the mechanical ventilation stack height were included to estimate the potential of increased concentrations very close to the source.

The CMAQ modelled concentrations were used as background concentrations for $NH_3$ and $PM_{2.5}$. Indeed, the concentrations calculated by CMAQ or other CTMs with a somewhat-coarse resolution are mostly representative of the background

conditions.

### 2.3.2    Emissions

The emissions in the regional modelling have been calculated with the SMT, based on national emissions, whereas the local modelling has used a combination of emission rates derived from measurements undertaken as part of this project (Leonard and Wiltshire, 2025) and in the absence of measured emissions the Simple Calculation of Atmospheric Impact Limits

(SCAIL) agricultural emission inventory (Hill et al., 2014) has been used.

As such local modelling has focused on five farms to reflect locations included in the measurement campaign. These farms have remained anonymous for the study. Details on the farms included in local modelling such as livestock type, number of sources, those that include measured or SCAIL emission inventories and mitigation have been detailed in Table 2.

The local dispersion modelling for all studied farms uses the same methodology, except for the development of the emission

rates which was unique to each farm depending on availability of activity and monitoring data from farms. However, farm activity and monitoring data were reviewed in a consistent approach across each farm with the final data used varying to reflect level of detail available. The further sub-sections detail the methodology adopted across all farms.

**Table 2: Farms included in local dispersion modelling.**

| Farm | Type of livestock | Sources | Measured or SCAIL sources | Mitigation |
|------|------|------|------|------|
| One | Pig | Two mechanically ventilated housing units with 4 fans each and 2 slurry lagoons | Measured at both housing units. SCAIL emission rate for slurry lagoon. | **Housing - ventilation scrubber** $NH_3$ 80% $NH_3$ reduction (SMT) $PM_{2.5}$ 60% $PM_{2.5}$ reduction |

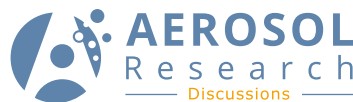

| | | | | (European Commission. Joint Research Centre., 2017) **Slurry lagoon** Floating cover 60% $NH_3$ (SMT) |
|---|---|---|---|---|
| Two | Pig | One naturally ventilated housing unit, 11 Mechanically ventilated housing units with 25 fans and 2 manure piles | SCAIL at naturally ventilated, 1 mechanically ventilated and 2 manure piles. Measured at 10 mechanically ventilated. | **Housing - ventilation scrubber** $NH_3$ 80% $NH_3$ reduction (SMT) $PM_{2.5}$ 60% $PM_{2.5}$ reduction (European Commission. Joint Research Centre., 2017) **Manure piles** Manure cover 60% $NH_3$ (SMT) |
| Three | Poultry, broilers | Eight mechanically ventilated housing units | Measured at 8 mechanically ventilated housing units | **Housing - ventilation scrubber** $NH_3$ 80% $NH_3$ reduction (SMT) $PM_{2.5}$ 35% $PM_{2.5}$ reduction (European Commission. Joint Research Centre., 2017) |
| Four | Poultry, broilers | Three mechanically ventilated housing units | Measured at 3 mechanically ventilated housing units | **Housing- ventilation scrubber** $NH_3$ 80% $NH_3$ reduction (SMT) $PM_{2.5}$ 35% $PM_{2.5}$ reduction (European Commission. Joint Research Centre., 2017) |
| Five | Dairy | Five naturally ventilated housing units, 1 manure pile, 1 yard, 1 slurry lagoon and 1 grazing area. | One measured naturally ventilated housing unit. Remaining sources used SCAIL. | **Grazing** Extend grazing period from 4 to 9 months (SMT). No % reduction applied to |



| | | | | pollutants, lower housing emissions achieved extending duration livestock are in pastures. |
|---|---|---|---|---|
| | | | | |


Detailed questionnaire, interview results and pollutant ($NH_3$ and $PM_{2.5}$) measurements collected from each farm in this study were reviewed to establish ADMS' source type representation such as point, volume and area and extent of time varying profile to apply. The primary emission data used in the modelling has used the same quality assurance protocol detailed within the measurement study (Leonard and Wiltshire, 2025), with monitoring data being processed into hourly averages to

reflect hourly meteorological limitations of ADMS. The measurement, questionnaire and interview results were used to establish existing emission profiles, any existing mitigation measures to lower $NH_3$ or $PM_{2.5}$ were reflected in the baseline. However, none of the mitigation measures recommended in this study (Jenkins and Wiltshire, 2024) were in place at farms (Leonard and Wiltshire, 2025). The order of preference for time varying emission profile development, with most preferred to least preferred below:

- Preferred emission profile - unique calculation for every hour in year

An emission rate (g/s) for every hour in a year is the most detailed emission input option in ADMS 6, as emission measurements at farms were undertaken for periods over 2022 and 2023 did not represent a full year of measured emissions from sources. As such the most detailed option available for each farm would be to develop an emission rate (g/s) for every hour in the animal cycle, then extrapolate this over a year based on reports of all the animal cycles in a year. There was only

sufficient monitoring and animal cycle data for each hour to have an emission rate at farm four (poultry). As there are only housing emission sources at farm four every source on this farm was based on an individually calculated emission rate for every hour in a year.

- 2nd emission profile preference – annual average emission rate for each hour in a day

The next level of detail available to develop time varying emission profiles at each farm was to calculate annual average

hourly emission rates (g/s) for the application of a diurnal profile in local modelling. This was applied to sources on farms one (pig), two (pig), three (poultry) and five (dairy) with measurement data. At pig farms one and two, this profile was applied to housing units with measurement data, but also to housing units based on the SCAIL emission inventory as the profile was considered relevant. At farm three (poultry) a diurnal profile based on annual average hourly emission rates (g/s) was applied to all housing units. The milking and loafing area on farm five was the only building with emission

measurements and the only building with a diurnal profile applied. Loafing areas are where cows on-lying, non-passageway, non-feeding spaces enable cows' freedom to express normal behaviour, such as grooming and heat expression. Grazing areas and housing for cattle that graze had two unique emission rates to reflect time of year grazing and housed.

- 3rd emission profile preference – constant emission rate for all hours in a year





The lowest level of detail is where no measurement or activity data was available to understand how annual emissions

should vary throughout the day and or year. In this situation annual emissions are divided by the number seconds in a year to derive a constant (g/s) for all hours in a year. No diurnal profile was applied to slurry and manure lagoons at farms one and two. At farm five (dairy) no diurnal profile was applied to the yard, slurry lagoon or manure piles.

Information on emission sources including dimensions, fan height, diameter, exit velocity were derived from farmer data requests and interviews. Housing temperature data was derived from either farm owned temperature sensors if available, or

from project monitoring equipment. Project measurements of $NH_3$ and $PM_{2.5}$ were processed to get either average $NH_3$ and $PM_{2.5}$ emission rates for each hour in an animal cycle or the entire measurement period. The processing of $NH_3$ and $PM_{2.5}$ measurement data are shown in Equation 1 and 2, respectively. Equations 1 and 2 are relevant for each individual hour in a flock cycle or period hourly average emissions.

$$ER_{NH3} = C_{NH3} \times Q \times R_{molecular} \times c_{mass} \quad (1)$$

$ER_{NH3}$ corresponds to the $NH_3$ emission rate (g/s), $C_{NH3}$ is the period or animal cycle hourly average $NH_3$ concentration (ppb), Q the volumetric flow rate ($m^3$/s) and $R_{molecular}$ is the ratio between the molecular weight and molecular volume and $c_{mass}$ the conversion constant ($10^6$).

$$ER_{PM2.5} = C_{PM2.5} \times Q \times c_{mass} \quad (2)$$

With $ER_{PM2.5}$ being the $PM_{2.5}$ emission rate (g/s), $C_{PM2.5}$ the period or animal cycle hourly average $PM_{2.5}$ concentration

($\mu g/m^3$), Q the volumetric flow rate ($m^3$/s) and $c_{mass}$ the conversion constant ($10^6$).

For instances where emission rate values could not be calculated, the SCAIL emission inventory was used. SCAIL emission rates are provided as $kg/m^2$ or kg per animal place per year, as such the area of sources and number of livestock were used in this equation to derive $NH_3$ and $PM_{10}$ kg/year. SCAIL emission rates are in $PM_{10}$ and this was converted into $PM_{2.5}$ by

looking at the ratio between $PM_{10}$ and $PM_{2.5}$ at Defra's Automatic Urban and Rural Network (AURN) rural background monitoring stations available at the UK AIR platform (DEFRA, 2024a) to derive a factor of 0.58. The measured emission rates were adjusted using Equation 3 for comparison with SCAIL annual emission (kg/year). This calculation assumes that the emission rate of one fan is representative of the concentration of the pollutant throughout the building and therefore can be scaled up using the building volume.

$$EF = ER \times V_{building} \times c_{mass} \times c_{time} \quad (3)$$

With EF the emission factor (kg/yr), ER the hourly average emission rate ($\mu g/(m^3.h)$), $c_{mass}$ the conversion constant ($10^9$) and $c_{time}$ the time conversion constant ($24 \times 365$).

- Mitigation scenario emission calculations



In the mitigation scenario, each emission source and associated percentage reduction from mitigation detailed in Table 2 were applied to emission rates (g/s). For example, acid scrubbers are applicable treatment of ventilated air at farm one animal
housing and the emission rate (g/s) is multiplied by 0.2 and 0.4 to reflect the proposed 80% and 60% reduction in $NH_3$ and $PM_{2.5}$, respectively.

## 3    Change in $PM_{2.5}$ concentrations

### 3.1    Regional Scale

#### 3.1.1    Evaluation of the historical simulation

The modelled concentrations have been evaluated in using the historical simulation in 2019. Only $PM_{2.5}$ measurement data for rural background sites with at least 75% data capture in the year are used to avoid bias. The observations were downloaded from the UK AIR platform. This represents a total of 48 stations. The CMAQ annual map and the comparison with the observations at the measurement sites are shown in Figure 3. The statistics used in this evaluation are described in Appendix C.

While the comparison shows a fair agreement in the correlation (r ~ 0.6), a clear underestimation in the modelled concentrations is calculated (mean bias (MB) ~ 5 µg/m$^3$; normalized mean bias (NMB) ~ -51%). This approximately 50% underestimation in the modelled $PM_{2.5}$ concentrations echoes the 50% homogenous increase in $NH_3$ emissions (and 60% decrease in $SO_2$ emissions) applied by Kelly et al. (2023) and Marais et al. (2023) in using a similar emissions inventory (NAEI for the year 2019) in their simulations to obtain a reasonable agreement in their calculated $PM_{2.5}$ concentrations with
their global CTM (r=0.66, NMB=-11%). However, it is worth noting a sensitivity simulation, by increasing our UK $NH_3$ emissions by 50% was also tested. Despite this large change in the 2019 $NH_3$ emission, no real improvement in the comparison with the observations was found (Fig. S1). This confirms the finding in Pommier et al. (2025) showing $NH_3$ is not 'limiting', thus $NH_3$ emissions changes will have a negligible on mitigating secondary inorganic aerosols (SIA) formation at regional scale. Kelly et al. (2023) also explained with $NH_3$ being in excess, the emissions scaling applied to $NH_3$
to resolve differences between top-down and bottom-up emissions estimates has only a limited effect on $NH_4$ and $PM_{2.5}$.
This might also suggest unrepresented atmospheric processes in the model between $NH_3$ and the $PM_{2.5}$ formation since this 50% increase in $NH_3$ emission leads to an overestimation of the modelled $NH_3$ concentrations (Pommier et al., 2025). For example, this could be a result of combined missing processes since the bi-directional $NH_3$ flux representation has not been implemented in this CMAQ simulation and this bidirectional treatment of $NH_3$ fluxes should improve the prediction of $NH_3$
(e.g. Pleim et al., 2019). It has been noted that assimilating satellite $NH_3$ observations help to improve the models' performance to calculate the surface SIA concentrations (e.g. Momeni et al., 2024). In addition, dry $PM_{2.5}$ concentrations has been used in the comparison and, without being the major contributor of these differences with the observations, the effect of



aerosol water on the mass closure of PM$_{2.5}$ can influence the value in the total PM$_{2.5}$ concentrations (AQEG, 2012; Kelly et al., 2023; Tsyro, 2005).

It is worth noting the main PM$_{2.5}$ component calculated by CMAQ for these stations is NO$_3$ (Tab. S1) and this composition spatially varies as shown on the maps (Fig. S2).

In the baseline 2019 simulation, a low Mean Relative Error (MRE ~ -0.5 %) has been calculated while the Root-Mean-Square Error (RMSE ~ 5 µg/m$^3$) and IOA (~0.4) are not fully satisfactory.

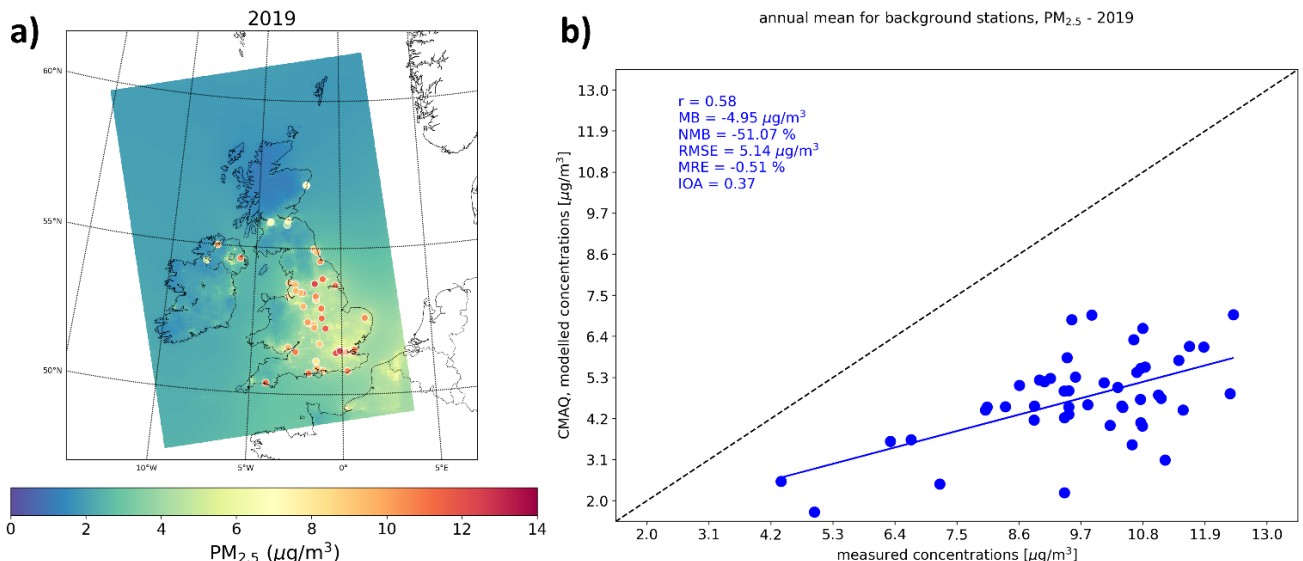

**Figure 3: a) Spatial distribution of annual mean PM$_{2.5}$ concentrations in µg/m$^3$ calculated by CMAQ at 10 km resolution in 2019.**
**The measured concentrations at the monitoring stations are shown with the coloured circles. b) Comparison between these annual measured concentrations with the modelled values in 2019. Only the background stations with a data capture higher than 75% are used. Insert values are the Pearson correlation coefficient (R), the mean bias (MB), the normalized mean bias (NMB), the mean relative error (MRE), the root-mean-square error (RMSE), and the index of agreement (IOA). The blue line represents the linear fit and dashed black line is the 1:1 slope.**

**3.1.2 Future changes**

Reductions in NH$_3$ emissions are effective at reducing NH$_3$ concentrations and its deposition at a regional scale (10 km × 10 km) as shown in Pommier et al. (2025) (e.g. up to 22% reduction in the high2030 scenario) but considerably less effective at reducing ammonium (NH$_4$) since the UK is characterized by an NH$_3$-rich chemical domain. This confirms the finding that the decrease in NH$_3$ emissions only has limited effects on mitigating SIA formation found by Ge et al. (2022) and that rural
areas are less sensitive to changes in NH$_3$ (Pan et al., 2024). Consequently, the PM$_{2.5}$ concentrations are only slightly impacted by the mitigation on agricultural activities implemented in our scenarios, as shown in Figure 4. Indeed, the reduction in the annual mean PM$_{2.5}$ concentrations is marginal for the three scenarios, since the largest calculated reduction is around 1.2%, 1.3% and 1.5% for the low2030, medium2030 and high2030 scenario, respectively; and the mean reduction is nearly null.



At the opposite, Ge et al. (2023) showed an important impact of the $NH_3$ emission reduction in $PM_{2.5}$ concentrations in the UK. The results in Ge et al. (2023) are not comparable with our study, since their analysis was based on a large decrease in the emissions, 4 times larger than our more ambitious mitigation (high2030) scenario. This difference in the assumption of the emissions' reduction, has a crucial impact on the atmospheric chemical regime and so changing the influence of $NH_3$ in the SIA formation.

Moreover, the scenarios have focused on mitigating $NH_3$ emissions, while targeting other secondary $PM_{2.5}$ precursors ($NO_x$ and $SO_x$) can be needed to effectively curb the $PM_{2.5}$ exposure (Marais et al., 2023; Pastorino et al., 2024).

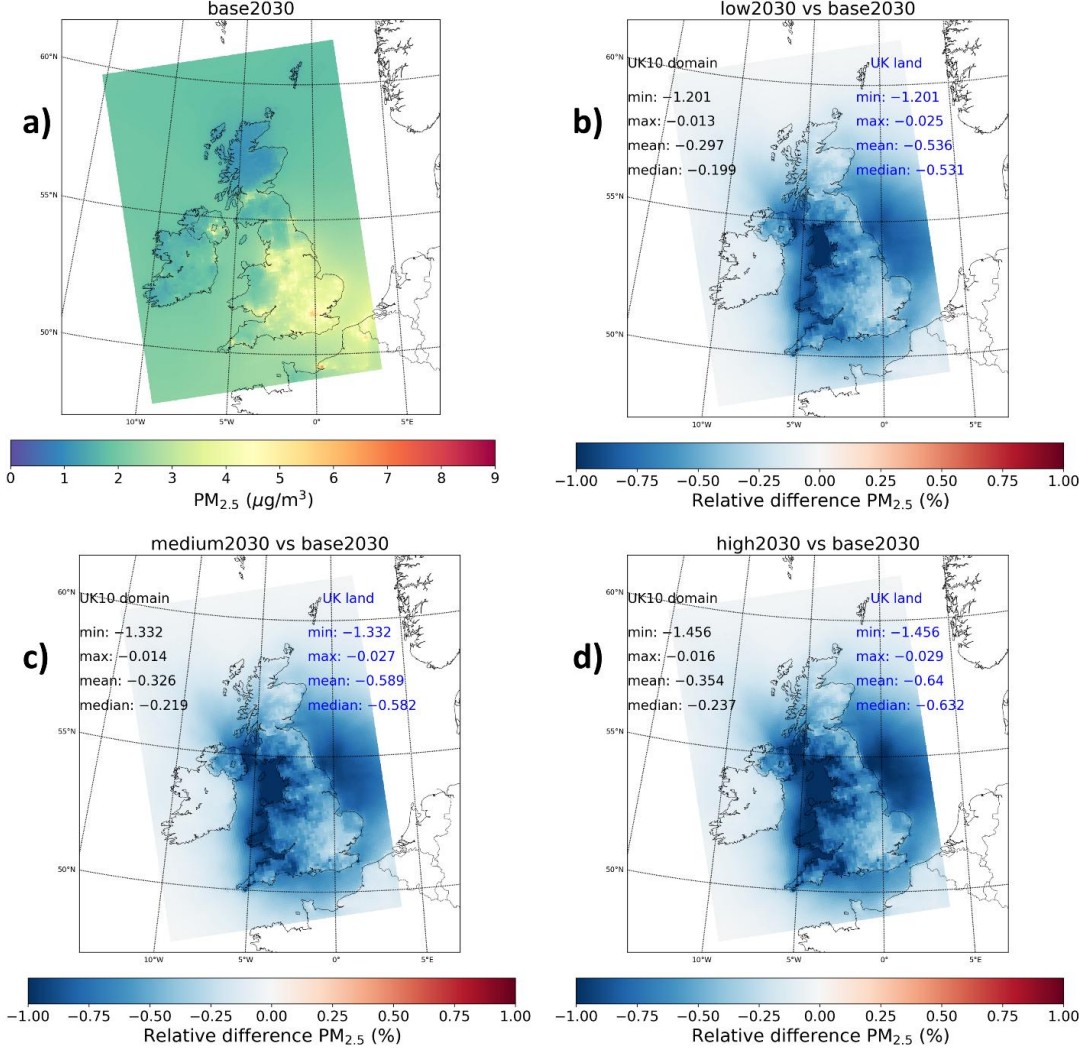

**Figure 4: a) Spatial distribution of annual mean $PM_{2.5}$ concentrations in µg/m3 calculated by CMAQ at 10 km resolution for the base2030 scenario. Relative difference of the same distribution with the low2030 (b), medium2030 (c) and high2030 (d) scenarios. The minimum, maximum, mean, and median relative difference values in the whole UK10 domain (in black) and for the UK land grid cells (blue) are provided. The relative difference is calculated as follow: ((scenario-base)/base) × 100%.**





## 3.2    Local scale: dispersion near the farms

Regional modelling has been used to estimate the contribution of agricultural $NH_3$ to the formation of secondary $PM_{2.5}$ at a regional scale, whereas local scale modelling has been used to investigate dispersion of $NH_3$ and $PM_{2.5}$ closer to farms (within 10km). This is a different modelling approach to the regional modelling that includes atmospheric chemistry to

estimate $PM_{2.5}$ through primary contributions and secondary formation, a non-steady state (reactive chemistry) option was reviewed in the local modelling, although secondary formation was lower than 1% of total $PM_{2.5}$ in the 10km study area and discounted from modelling. However, both modelling approaches are linked since the regional modelled concentrations have been used to define the background concentrations.

As detailed in Section 2.1, low to high mitigation refers to mitigation uptake by number of farms, but local modelling

focuses on five specific farms and variable uptake values are not relevant. Instead, consistent $NH_3$ impact values (percentage reduction) were adopted between regional and local modelling, with $PM_{2.5}$ impact values (percentage reductions) derived separately through best practice agricultural guidance (European Commission. Joint Research Centre., 2017). Mitigation measures were assessed in the local modelling scenario to gauge the maximum potential benefit on pollutant concentrations in local vicinity of farms.

Figure 5 represents study farm's contributions of $NH_3$ and primary $PM_{2.5}$ under existing farm operations (base2030), under the mitigation scenario and their differences. The mitigation scenario for the local modelling features all measures from the low2030, medium2030 and high2030 scenarios, whereas regional modelling represented increasing percentage uptake nationally from low to high scenarios, local modelling implemented mitigation measures relevant for specific farms. As reminder, the mitigation measures for each farm are described in Table 2.

Across the existing and mitigation scenarios the greatest distance for concentrations of $NH_3$ and $PM_{2.5}$ to reach 10% of the maximum is 700 metres (Fig. 5a). The distance at which concentrations reach 10% of the maximum varies depending on many local scale dispersion parameters at the farm and meteorology, such as air flow release rate (m/s), temperature (℃), wind speed (m/s) and direction (°) and impact of building downwash.

50% of air pollutant concentrations from farm two are dispersed at a closer distance (100m) than other farms due to an air

flow rate of 5.1 m/s, whereas farms one, four and five have a flow rate ranging between 7 and 11.5 m/s which contributes to the plume grounding at a closer distance to farm two.

It is worth noting that the mitigation scenario solely impacts the distance of spread of the pollutants for the farm three, while the distances where the 50% of $NH_3$ and primary $PM_{2.5}$ concentrations are dispersed; and the distances where 10% of their maximum concentrations are found are identical for the other farms (Figs. 5b & c).




Figure 5: a) Farm's contributions of NH₃ and primary PM₂.₅ given as a distance in meters where the concentration if 50% or 10% of maximum for the base2030 scenario (a) and the mitigation scenario (b).



The difference in concentrations between the mitigation and base2030 scenarios are presented in Table 3 in terms of maximum concentration in a 10km$^2$ area, and maximum concentration for sensitive receptors. Table 3 shows that within 1km

of farms included in this study there can be reductions between 25 and 80% of total NH$_3$ concentrations and 4 and 60% reductions of PM$_{2.5}$.

The biggest reductions in pollutant concentrations occur at farm one and two, which are pig farms and the abatement measure with the biggest benefit is an acid scrubber used to reduce emissions from housing and as shown in Table 2 is estimated to achieve an 80% reduction in NH$_3$ and 60% reduction in PM$_{2.5}$ emissions.

The only other relevant mitigation measure included at farms one and two would be to provide a cover over open manure and or slurry lagoons, however this has a smaller 60% reduction of only NH$_3$ emissions and will have a smaller impact on NH$_3$ concentrations than the acid scrubber. While acid scrubbers and manure/slurry covers are included in modelling of estimated concentration the biggest will come from acid scrubbers.

**Table 3: Percent difference in concentrations between base2030 and mitigation scenarios.**

| Farm | Reduction in max concentration in 10km$^2$ study area (µg/m$^3$) | | Reduction in max concentration for sensitive receptors (µg/m$^3$) | |
|---|---|---|---|---|
| | PM$_{2.5}$ | NH$_3$ | PM$_{2.5}$ | NH$_3$ |
| Farm one (pig) | -60% | -79% | -60% | -80% |
| Farm two (pig) | -60% | -63% | -60% | -64% |
| Farm three (poultry, broilers) | -13% | -25% | -31% | -71% |
| Farm four (poultry, broilers) | -35% | -80% | -34% | -80% |
| Farm five (dairy) | -4% | -43% | -7% | -33% |

## 4    Discussion

The design of the emission scenarios was based on the views of farmers, advisers, academics, and representatives from relevant sectors, capturing diverse perspectives and making the uptake scenarios grounded in real-world practices and

challenges. This approach also considered the actual barriers and incentives that farmers experience, leading to realistic projections of mitigation measure uptake. Using multiple engagement tools (online surveys, focus groups, and one-on-one interviews) also enabled the gathering of in-depth, well-rounded data, providing a nuanced understanding of the factors influencing uptake. However, it is worth noting that the future uptake projections did not account for potential changes in legislation, which could significantly impact the adoption of mitigation measures. This limits the ability to predict uptake

under different regulatory environments. Moreover, the method has not differentiated uptake scenarios between different parts of the UK due to a lack of data, potentially overlooking regional variations in farming practices, environmental conditions, or economic incentives. The study has also relied on subjective feedback, which can vary widely between





individuals or groups. This can introduce bias in determining which measures are positively or negatively received, potentially affecting the estimated uptake rates.

Although CMAQ is state of the art and widely used in scientific research and policy development, the models also has uncertainties. The analysis presented in this study rely on the accuracy of the simulation which is subject to any uncertainties in the model's specific parameterization of atmospheric processes, as well as uncertainties in the emission inventory and meteorology input. It has been shown that CMAQ does not perfectly model the interactions between $NH_3$ emissions and the $PM_{2.5}$ formation which can be explained by the local processes causing the majority of $NH_3$ to be dispersed near the studied

farms as highlighted by ADMS results showing a 90% decrease in concentrations within 700 metres of farms. This study confirms the findings from Pan et al. (2024) arguing for more collocated aerosol and precursor observations for better characterization of SIA formation.

The limited impact of the mitigation measures at a regional scale, which mainly target the $NH_3$ emissions, on $PM_{2.5}$ concentrations is due to an $NH_3$-rich atmosphere in the UK and highlights that other precursor of these $PM_{2.5}$ and the primary

$PM_{2.5}$ emissions need to be tackled. This also highlights that exposure on secondary $PM_{2.5}$ near the farms needs also to be investigated while most air quality studies focus on total $PM_{2.5}$ concentrations. ADMS has showed that the majority (90%) of secondary $PM_{2.5}$ precursor $NH_3$ emissions and primary $PM_{2.5}$ is dispersed within 700 metres of farms. This supports conclusions from CMAQ of little impact on a regional scale as most relevant exposure is beyond 700 metres of farms. An area of further work is recommended to review the impact of mitigation measures on primary and secondary $PM_{2.5}$ at

relevant human health exposure within 1 to 10km of farms, as national exposure weights impact towards locations where the majority of primary pollution has dispersed.

Limitations in the local modelling include uncertainties associated with project measurement data and associated activity data from farms. The project measurement study (Leonard and Wiltshire, 2025) should be referenced for the full suite of limitations associated with project measurement data, however the main aspects that affect emission rates developed for

local modelling includes representativeness of measurement location for entire housing unit, that measurements did not span an entire animal cycle at farms one, two and five. Regarding representativeness of measurements, at farms three and five housing air was sampled with a multiplexer, a device that samples air from multiple locations, whereas measurements at other farms only sampled air from one location. As such a limitation of emission rates used in modelling is the assumption emission rates are representative for the entire animal housing unit. Measurement data did not span entire animal lifecycles

at farms one, two and five and as such the project measurement data and housing emissions rates are limited in how representative they are of each animal lifecycle. Further to this, farms one, two and five did not record animals in each housing unit for each day of the measurement period and over the animal lifecycle, instead assumptions were made on the total number of animals apportioned to each housing unit. Consequently, there is uncertainty regarding animal numbers in each housing unit and extrapolations made for the annual animal places at farms one, two and five. Whilst farms two and

three had measurements for the entire animal cycle, like farms one and two measured fan flow rates were not available during the measurement period and ventilation manufacturer's records were used to develop air flow rates. Whilst there are





limitations in data used, replacing emission and flow rate assumptions is unlikely to alter that the majority of pollution is grounded in the nearfield (<1km) of farms, since agricultural sources are emitted from lower heights (<6m) and have low air flow rates relative to other sources such as engine exhausts.

## 5    Conclusions

This study highlights the complex interactions between $NH_3$ emissions from farming activities and $PM_{2.5}$ formation in the UK, with a focus on dairy, pig, and poultry sectors. Using both CMAQ model for regional-scale analysis and ADMS for local-scale dispersion, this work has evaluated the impact of mitigation measures under various uptake scenarios on reducing emissions, especially on $NH_3$. Although emission reductions, particularly in $NH_3$, were predicted under high uptake scenario, these changes did not translate into significant reductions in regional-scale $PM_{2.5}$ concentrations, with a maximum decrease of only 1.5%. This outcome is attributed to the $NH_3$-rich atmosphere, which diminishes the effect of $NH_3$ reductions on $PM_{2.5}$ mitigation.

The findings also reveal discrepancies between CMAQ model concentrations and ground-based measurements, suggesting that key atmospheric processes influencing $PM_{2.5}$ formation may not be fully represented in the model, leading to an underestimation of $PM_{2.5}$ concentrations by approximately 50%. ADMS results further show that $NH_3$ is rapidly dispersed near the farms, indicating a limited role of these emissions in the formation of $PM_{2.5}$ locally. The study has emphasized the need for integrated modelling approaches and better characterization of SIA formation, as well as the importance of addressing the primary $PM_{2.5}$ and other $PM_{2.5}$ precursors beyond $NH_3$ to achieve effective air quality improvements.

Overall, this suggested limited impact on potential $NH_3$-focused mitigation strategies on $PM_{2.5}$ concentrations underscores the necessity of exploring additional emission control measures targeting other precursors and primary $PM_{2.5}$ emissions from the farming sector. Indeed, further work is recommended to review the national benefit of mitigation on primary $PM_{2.5}$ emissions, however benefits of mitigation are likely to be localised on $PM_{2.5}$ as demonstrated by ADMS modelling. Future research should also focus on primary and secondary $PM_{2.5}$ exposure separately near farms, as current air quality studies predominantly assess total $PM_{2.5}$ concentrations, and further work is required to understand the impact of secondary $PM_{2.5}$ on health. This work advocates for a more holistic approach to modelling and mitigation to better inform policies aimed at improving air quality in agricultural regions.

The study has looked at regional exposure to $PM_{2.5}$ from agricultural sources in CMAQ, whereas ADMS has shown that the majority (90%) of emission are dispersed within 700m of farms. As the UK population is concentrated in urban areas a substantial distance from farms, further work could explore the health benefit of mitigation on communities in the local vicinity of farms (from 1 to 10km).



**Appendix -A**

Table A1 summarises the measures and the uptake rates for each of the three scenarios for the regional modelling. These values are additional to uptake of measures already included in emissions from NAEI.

The uptake scenarios were developed through stakeholder engagement with farmers and stakeholders (i.e. farm advisers, academics and farmer representatives). Each scenario includes all 19 mitigation measures, however with varying percentages of uptake.

The uptake rates were unique to each mitigation measure in each sector and were reflective of feedback received through engagement activities. The engagement activities included an online survey, focus groups and one-to-one interviews with participants from the dairy, pig and poultry sectors and those in other sectors which utilise manure or slurry. A total of 161 people took part in the activities. Full results and methodology are detailed in Jenkins and Wiltshire (2024)

Discussions in these activities were centred around understanding the current level of uptake and the benefits and barriers associated with the mitigation measures to determine a potential future uptake. If a mitigation measure was received positively, it was estimated to have a higher uptake compared to measures that were received negatively by participants. This was determined in the final level of uptake for each scenario. The future uptake did not take account of any potential changes to legislation that may have an impact as this information is not known, additionally there were no different uptakes for each part of the UK due to a lack of data.

**Table A1. A summary of the measures and uptake rates used in each of the three scenarios modelled for this study.**

| Sector | Measure | Uptake (%) | | |
|--------|---------|------|--------|------|
| | | Low | Medium | High |
| Poultry | Planting trees near livestock housing | 75 | 80 | 85 |
| Poultry | Installing air scrubbers to filter pollutants | 0 | 1.5 | 3 |
| Poultry | Covering a solid manure heap with a sheet | 80 | 85 | 90 |
| Poultry | Amending diet to better match the nitrogen content to livestock need | 97 | 98 | 99 |
| Poultry | In-house poultry manure drying | 10 | 12.5 | 15 |
| Poultry | Increased litter removal (e.g. by belt removal) | 50 | 52.5 | 55 |
| Pig | Planting trees near livestock housing | 42 | 47.5 | 53 |
| Pig | Trailing shoe | 19 | 22.5 | 26 |
| Pig | Trailing hose | 10 | 13 | 16 |
| Pig | Using slurry bags | 2 | 3 | 4 |
| Pig | Acidification of slurry in underfloor storage tanks in housing units | 1 | 2 | 3 |
| Pig | Installing air scrubbers to filter pollutants | 0 | 1.5 | 3 |



| | | | | |
|---|---|---|---|---|
| Pig | Shallow injection - open slot | 19 | 21.5 | 24 |
| Pig | Permeable floating cover (e.g. chopped straw) on slurry store | 8 | 13 | 13 |
| Pig | Amending diet to better match the nitrogen content to livestock need | 97 | 98 | 99 |
| Pig | Increasing bedding in housing (e.g. straw) | 31 | 36 | 37 |
| Pig | Vacuum/flushing system for slurry removal from pits under slatted flooring | 12 | 14 | 16 |
| Pig | Impermeable floating sheet on slurry store | 5 | 10 | 18 |
| Pig | Using a fixed solid cover on slurry stores | 15 | 17.5 | 20 |
| Pig | Improving pen design to keep solid parts of the floor as clean as possible | 20 | 25 | 27 |
| Pig | Covering a solid manure heap with a sheet | 5 | 7.5 | 10 |
| Pig | Using automatic or robotic scrapers | 30 | 35 | 36 |
| Dairy | Covering a solid manure heap with a sheet | 5 | 7.5 | 10 |
| Dairy | Planting trees near livestock housing | 42 | 47.5 | 53 |
| Dairy | Using trailing shoe | 18 | 24 | 30 |
| Dairy | Using trailing hose | 35 | 40 | 45 |
| Dairy | Acidification of slurry in underfloor storage tanks in housing units | 0 | 1.5 | 3 |
| Dairy | Shallow Injection | 13 | 15.5 | 18 |
| Dairy | Using robotic scrapers (e.g. Lely Sphere) | 7.5 | 10 | 12.5 |
| Dairy | Permeable floating cover (e.g. chopped straw) on slurry store | 8 | 13 | 18 |
| Dairy | Amending diet to better match the nitrogen content to livestock need | 95 | 97 | 99 |
| Dairy | Increasing washing in yards/parlours from once to twice a day | 10 | 15 | 20 |
| Dairy | Increasing scraping in yards/parlours from once to twice a day | 40 | 41 | 43 |
| Dairy | Increasing bedding in housing units (e.g. straw) | 17 | 18 | 20 |
| Dairy | Impermeable floating sheet on slurry store | 5 | 10 | 15 |
| Dairy | Using a fixed solid cover on slurry stores | 41 | 43.5 | 46 |
| Dairy | Extending the grazing season | 74 | 79.5 | 85 |
| Dairy | Using automatic scrapers | 25 | 27.5 | 30 |






## Appendix - B

Table B1 presents the practices that reduce ammonia emissions that were modelled in this study, along with a brief description on how it reduces ammonia.

**Table B1. Practices that reduce ammonia emissions, with a short description of how they reduce emissions.**

| | Practices that reduce ammonia emissions | How does it reduce ammonia emissions? |
|---|---|---|
| Housing | Extending the grazing season | Grazing animals urinate directly on the grass. The urine then infiltrates, reducing the exposure to air. |
| | Increasing bedding material (e.g. straw, sand) | Increasing the amount of bedding helps to absorb more urine, reducing exposure to air. |
| | Increasing washing and scraping in yards areas | Scraping urine, slurry and manure into a covered store reduces the exposure to the air and the reaction to produce ammonia. |
| | Increasing cleaning by using automatic or robotic scrapers | As above, more frequent cleaning reducing the exposure to air. |
| | Acidification of slurry (usually in housing with an under-floor slurry pit) | Lowering the pH, by adding an acid such as sulphuric acid, decreases emission. |
| | Amending livestock diet to match N content to the amount of growth | Matching feed to the required amount for growth reduces the excretion of excess N, some of which will be emitted as ammonia. |
| | Planting tree shelter belts near livestock housing | Emissions are dispersed and/or taken up by the tree foliage. |
| | Moving livestock housing away from sensitive sites (e.g. SSSIs) | A drastic option, but effective because ammonia is deposited near the place of emission. This measure moves the sources of ammonia away from sites sensitive to ammonia depositions. |
| | Reducing stocking densities near sensitive sites (e.g. SSSIs) | Moves the sources of ammonia away from sites sensitive to ammonia depositions. |
| | Installing air scrubbers to filter pollutants | Fitted to housing units to remove ammonia. |
| | Increased checking of water structures to reduce leaks | More ammonia is emitted if bedding is wet |
| | Increasing litter removal (e.g. by belt removal) | For layers, collecting and removing manure to a covered store, reducing exposure to air. |
| Stora | Slurry bags | Creates a physical barrier between the manure/slurry and the air. |
| | Covering stores with a fixed solid cover | |



| | |
|---|---|
| Covering stores with an impermeable floating sheet | |
| Using a permeable floating cover (chopped straw) | |
| Covering a manure heap on permeable ground | |
| Trailing hose | Applies slurry in narrow bands at grass level, reducing the surface area, helping quicker infiltration and reducing exposure to air. |
| Trailing shoe | Applies slurry in narrow bands at soil level, reducing the surface area, helping quicker infiltration, reducing the exposure to air. |
| Shallow injection | Injecting slurry into the ground, helping quicker infiltration and reducing exposure to air. |









**Appendix - C**

Statistics used for the evaluation of the air quality simulation with CMAQ. In the following notations, M and O refer, respectively, to the model and the observations data. N is the number of the observation data set.

**Pearson relation coefficient (r):** The ideal score of these parameters is 1. It is an unitless variable.

**Mean bias (MB):** The ideal score of this parameter is 0. The unit of this variable is the as the pollutant concentration ($\mu g/m^3$). The MB provides information about the absolute bias of the model, with negative values indicating underestimation and positive values indicating overestimation by the model.

$$MB = \frac{\sum_{i=1}^{N}(M_i - O_i)}{N}$$


**Normalised mean bias (NMB):** The ideal score of this parameter is 0 and the unit of the variable is in percent. The NMB represents the model bias relative to the reference.

$$NMB = \frac{\sum_{i=1}^{N}(M_i - O_i)}{\sum_{i=1}^{N} O_i} \times 100\%$$

**Root-mean-square error (RMSE):** The ideal score of this parameter is 0. The unit of this variable is the as the pollutant
concentration ($\mu g/m^3$). The RMSE considers error compensation due to opposite sign differences and encapsulates the average error produced by the model.

$$RMSE = \sqrt{\frac{\sum_{i=1}^{N}(M_i - O_i)^2}{N}}$$

**Mean Relative Error (MRE):** The ideal score of this parameter is 0. The unit of this variable is the as the pollutant concentration ($\mu g/m^3$). The MRE is the mean ratio of difference between the model values and the observations, on the
observations.

$$MRE = \frac{1}{N}\sum_{i=1}^{N}\frac{M_i - O_i}{O_i}$$

**Index of Agreement (IOA):** The agreement value of 1 indicates a perfect match, and 0 indicates no agreement at all. It is an unitless variable.

$$IOA = 1 - \frac{\sum_{i=1}^{N}(M_i - O_i)^2}{\sum_{i=1}^{N}(|M_i - \bar{O}| + |O_i - \bar{O}|)^2}$$




**Code availability:**

The CMAQ model is freely provided by the US EPA: https://zenodo.org/record/7218076. The WRF model is freely available thanks to NCAR on https://github.com/wrf-model/WRF/tree/release-v4.5. The ADMS model is distributed under license by CERC: https://www.cerc.co.uk/environmental-software/ADMS-model.html.

**Data availability:**

Primary data from the regional, local modelling and emission measurements has been used in-combination with secondary data in this assessment. All data requests should be submitted to the corresponding author for consideration. Access to anonymised data may be granted following review.

**Author contribution:**

**MP**: Conceptualisation (equal), Data curation (equal), Formal analysis (equal), Investigation (equal), Methodology (equal), Project Administration (equal), Resources (lead) validation (equal), visualisation (lead), Writing – Original draft (lead), Supervision (lead). **RB**: Conceptualisation (equal), Data curation (equal), Formal analysis (equal), Investigation (equal), Methodology (equal), Project Administration (supporting), validation (equal), visualisation (supporting), Writing – Original draft (supporting). **JB:** Data curation (supporting), Investigation (supporting), Methodology (supporting). **BJ:** Methodology (supporting), Writing – Original draft (supporting). **JR:** Data curation (supporting), Formal analysis (supporting). **LR:** Methodology (supporting), Writing – Original draft (supporting). **OB:** Data curation (supporting), Formal analysis (supporting), Investigation (supporting), Methodology (supporting), Writing – Original draft (supporting). **OM:** Data curation (supporting), Formal analysis (supporting), Investigation (supporting), Methodology (supporting). **AS:** Data curation (supporting), Formal analysis (supporting), Investigation (supporting), Methodology (supporting).

**Competing interests:**

All authors were employed by the company Ricardo Energy & Environment. All authors declare that the research was conducted in the absence of any commercial or financial relationships that could be construed as a potential conflict of interest.

**Acknowledgements:**

The authors would like to thank the AURN measurements which are freely available on https://uk-air.defra.gov.uk/networks/network-info?view=aurn (access on 01 August 2025).










**Financial support**:

This project was funded by the National Institute for Health and Care Research (NIHR) AIM-HEALTH programme under award number NIHR 129449.

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
