# Peer review of "Impact of agricultural interventions on ammonia emissions and on PM2.5 concentrations in the UK: a local and regional modelling study"

_Aerosol Research, 2025_

## Author Comment (AC1)

We would like to thank the reviewer for his reading and his comments. We have written our replies to the comments in blue.

Please also note we have deleted the reference Cowie, H.: AIM-HEALTH: Effectiveness of agricultural interventions to minimise the health impacts of air pollution, NIHR, in press, 2025. While the study (Cowie et al) is accepted for publication, it still has not been published yet and there is no doi for the citation.

We have also amended the text in the manuscript and provided more results, shown as new figures in the supplement.

This manuscript presents different ammonia emission scenarios' impact on PM2.5 mitigation through both a chemical-transport model and a dispersion model. A major revision is recommended before acceptance.

**Major comments:**

Please add a paragraph to summarize ammonia emission sectors in the UK, e.g., the percentage of NH3 emissions from housed dairy, pigs, and poultry. Are there any noticeable industrial and vehicle NH3 emissions near the selected farms?

Concerning the NH3 emissions sectors we have added the following sentence in bold:

"Indeed, the most recent figure from the UK National Atmospheric Emissions Inventory (NAEI) shows that agriculture accounted for nearly 87% of total ammonia emissions in 2023 (NAEI, 2025). Direct soil emissions account for 52.7% of total NH3 emissions, followed by cattle at 25.9%, waste at 9.5%, other livestock at 4.8%, poultry at 3.7%, and combustion and production processes at 3.4%."

For the 2nd part of the question, ammonia mainly comes from agriculture in our studied local modelling which is the farm itself. There is no major roads or industries near the farms and so NH3 emissions from vehicles wouldn't be a significant source.

Thus, we have had the information (in bold) in the following sentence:

"Each farm, situated in a different region of the UK (Fig. 1), away from major roads and industrial areas, had a 15km-by-15km points grid centred at the farm with a 100m resolution."

The changes in emissions between different uptake scenarios are relatively small - low2030 is 8.5% and medium2030 is 9.3%. It is redundant to have 3 scenarios with such small emission differences.

One of the main novel aspects is developing realistic scenarios based on input from a range of stakeholders etc. In other words, rather than looking at a certain percentage reduction in  $NH_3$ , these estimates are grounded in specialist input from farmers, academia, and advisers.

Thus, despite unrealistic large reductions, the focus of the study was to assess the impact of realistic implementation of measures (even with small total emission reduction) on pollution.

We have added this sentence in Section 2.1:

"The overall aim was to assess realistic implementation of specific mitigation measures."

What is the temporal resolution of the model? Ammonia emissions in winter are much lower than in summer. Wintertime NH3 control is likely to be more effective than summertime. Please comment on the seasonality of ammonia emissions and PM2.5 compositions instead of using the annual mean.

Concerning the temporal resolution of the model we have added the following information (in bold) in Section 2.2.1: "The CMAQ model, calculating the pollutants' concentrations and depositions at an hourly resolution, ..."

To detail the seasonality in NH3 emissions and changes in PM2.5 concentrations, we have added this following information:

**- Section 2.2.2:**

"It is noteworthy that the UK NH3 emissions are mainly dominated by the February-April period as shown in Figure S1 and in Hellsten et al. (2007). Marais et al. (2021) reported an additional July peak associated with dairy cattle farming, based on satellite observations, alongside the spring peak. In contrast, the Emissions Database for Global Atmospheric Research (EDGAR) applies a uniform temporal profile for agricultural NH3 emissions in the UK in its latest inventory version (EDGARv8.1: <a href="https://edgar.jrc.ec.europa.eu/dataset\_ap81#plm">https://edgar.jrc.ec.europa.eu/dataset\_ap81#plm</a>, last access on 02.11.2025)."

**- Section 3.1.2:**

"It is also worth noting that the impact of the mitigation measures, even limited, varies by months, showing a larger relative change in May-July (only up to -3.4%) in the example of the high2030 scenario in Figure S5. These months do not correspond to the maximum in the emitted NH3 in the modelling as shown in Figure S1. This suggests also an impact of the atmospheric chemistry in the change in PM2.5 concentrations."

**- Conclusions:**

"Although the study primarily addresses annual estimates, further investigations at finer temporal resolutions (e.g., daily, monthly) could yield deeper insights into exposure impacts."

**The new figures are:**

Figure S1. Monthly temporal profile for  $NH_3$  emissions in agriculture sector in the UK.

Figure S5. Relative difference of the  $PM_{2.5}$  concentration with high2030 scenario compared to base2030 for each month. The minimum, maximum, mean, and median relative difference values in the whole UK10 domain (in black) and for the UK land grid cells (blue) are provided. The relative difference is calculated as follow: ((high2030-base2030)/base2030)  $\times$  100%.

**And references:**

- Hellsten, S., Dragosits, U., Place, C.J. et al. Modelling Seasonal Dynamics from Temporal Variation in Agricultural Practices in the UK Ammonia Emission Inventory. Water Air Soil Pollut: Focus 7, 3–13, https://doi.org/10.1007/s11267-006-9087-5, 2007.
- Marais, E. A., Pandey, A. K., Van Damme, M., Clarisse, L., Coheur, P.-F., Shephard, M. W., UK ammonia emissions estimated with satellite observations and GEOS-Chem, Journal of Geophysical Research: Atmospheres, 126, e2021JD035237. https://doi.org/10.1029/2021JD035237, 2021.

What is the interannual variability between 2019 and 2030? Are there any warming trends during the studied time period? If so, please comment on the impact on ammonia volatilization. Also, please comment on the contribution from transport and deposition changes.

The study has focused on the changes in future emissions and not on a change on meteorology, so no climate projection has been calculated and used.

The following sentence in lines 190-191 in the original manuscript mentioned this point: "The future scenarios solely focused on change in emissions and no climate projection has been undertaken." Thus, there is no analysis on the impact of meteorology on the volatilization.

We have added this sentence to clarify this point:

"Consequently, there is no analysis on changes in meteorological conditions."

Concerning the question on transport and deposition, we did not undertake a source apportionment study and the complementary study already focused on nitrogen deposition. There is already this sentence in lines 366-368 (beginning of Section 3.1.2):

"Reductions in  $NH_3$  emissions are effective at reducing  $NH_3$  concentrations and its deposition at a regional scale (10 km  $\times$  10 km) as shown in Pommier et al. (2025) (e.g. up to 22% reduction in the high2030 scenario) but considerably less effective at reducing ammonium ( $NH_4$ ) since the UK is characterized by an  $NH_3$ -rich chemical domain"

**Are NOx emissions underestimated? Any sensitivity test with NOx emissions?**

The evaluation of the NO2 concentrations modelled by CMAQ show a good agreement with the observations downloaded from the UK AIR platform as shown below, highlighting a good estimate in NOx emissions. However, no sensitivity simulation on NOx emissions was undertaken.

**We have added this sentence and figure:**

"In addition, the analysis on  $NO_2$  concentrations highlights a good estimate in  $NO_x$  emissions since a reasonable underestimation is found (~25.3%, 4.3  $\mu$ g/m³), with a good correlation (0.71) and IOA (0.78) (Figure S4)."

Figure S4. Spatial distribution of annual mean  $NO_2$  concentrations in  $\mu g/m^3$  calculated by CMAQ at 10 km resolution in 2019. The measured concentrations at the monitoring stations are shown with the coloured circles. b) Comparison between these annual measured concentrations with the modelled values in 2019. Only the background stations with a data capture higher than 75% are used. Insert values are the Pearson correlation coefficient (R), the mean bias (MB), the normalized mean bias (NMB), the mean relative error (MRE), the root-mean-square error (RMSE), and the index of agreement (IOA). The blue line represents the linear fit and dashed black line is the 1:1 slope.

Please clarify how meteorological differences between 2019 and 2030 are taken into consideration for the decay distance comparison.

As mentioned in our reply to a previous comment, no change in meteorology has been applied in this study.

Additionally, I suggest including a figure to display the concentration maps generated by the dispersion model for Farm 3.

The following sentence (in bold) and figure have been added:

"It is worth noting that the mitigation scenario solely impacts the distance of spread of the pollutants for the farm three, while the distances where the 50% of NH3 and primary PM2.5 concentrations are dispersed; and the distances where 10% of their maximum concentrations are found are identical for the other farms (Figs. 5b & c). However, as illustrated in Figure S6, farm three did not contribute to PM2.5, and the NH3 concentration remained highly localized around the farm."

Figure S6. Distribution of NH3 (a) and PM2.5 (b) surface concentrations in  $\mu g/m^3$  at farm three for the basse2030 scenario.

**Minor comments:**

Tabel S1: no SO42-?

Yes, it is correct. Thank you for spotting this mistake. SO4 has been forgotten in this table and now added.

---

## Author Comment (AC2)

We would like to thank the reviewer for his careful read, and we appreciate the time he took to provide his constructive feedback. We have replied below to his comments in blue.

We have also amended the text in the manuscript and provided more results, shown as new figures in the supplement.

Please also note we have deleted the reference Cowie, H.: AIM-HEALTH: Effectiveness of agricultural interventions to minimise the health impacts of air pollution, NIHR, in press, 2025. While the study (Cowie et al) is accepted for publication, it still has not been published yet and there is no doi for the citation.

The area of research encompassed by the title of this paper is of global interest and importance. PM2.5 is the air pollutant with the greatest human health burden, and ammonia emissions contribute to a significant proportion of PM2.5 composition, but are proving hard to abate. It is of interest to scientists and policy-makers to determine the extent to which NH3 emissions – from agriculture in particular – might be feasibly reduced, and the effect of these emissions reductions on PM2.5 concentrations.

Unfortunately, however, despite the important research context, this paper ultimately contains little useful new data or depth of insight. The data also appear to be subject to much uncertainty.

The paper is also not helped by writing that, in my opinion, is not of sufficiently good quality for a journal paper. There is much poor or confusing grammar, and long-winded ways of saying things. Scientifically, I found it hard to follow a lot of the description of the methodology, and I also found it very difficult to extract the points being made from a lot of the text in Section 3.2.

The revised version of the manuscript addresses the grammar issues, simplifies overly complex phrasing, and ensures that the key points are clearly articulated.

The detailed answers are provided in the rest of this document.

There is also a question of prior publication of a lot of the work. Scientifically, the paper reports independent applications of an atmospheric chemistry transport model (CMAQ) and of a local dispersion model (ADMS), to simulate, respectively, annual mean PM2.5 over the UK and rate of PM2.5 concentration fall-off from 5 individual farms. In both cases, model runs are performed using baseline emissions (for 2030) and scenarios corresponding to low, medium and high NH3 emission mitigation measures being applied to NH3 sources on farms nationally, or to the 5 individual farms specifically. The most novel part of the work overall is the methodology used to derive information about potential extent of uptake of various possible NH3 emission mitigation methods and the conversion of this qualitative scenario data into quantitative changes in actual NH3 emissions from farms. (According to what is written in the present paper, that part of the study involved questionnaires and discussions with farmers and other stakeholders, and use of emissions modelling tools.) However, all the scenario and emissions development is contained in other publications: Jenkin & Wiltshire, and Leonard & Wiltshire. The current paper uses these previously derived emissions and emission factors directly.

Yes, it is correct. This paper written by Jenkin & Wilshire is now available online. Please see the full reference here:

Jenkins, B.; Wiltshire, J. Farmer perceptions of the benefits and barriers to ammonia mitigation measures. Preprints 2025, 2025082071. <a href="https://doi.org/10.20944/preprints202508.2071.v1">https://doi.org/10.20944/preprints202508.2071.v1</a>

Similarly, the AQ measurements paper is also available on:

Leonard, A.; Wiltshire, J. Agricultural Emissions Measurements from Five English Farms. Preprints 2025, 2025082187. https://doi.org/10.20944/preprints202508.2187.v1

The references have been updated to the revised manuscript.

It is also worth noting that one of the novel aspects of this study was to develop realistic scenarios based on input from a range of stakeholders, academia etc., rather than looking at a certain unrealistic percentage in emission reductions.

We have added this sentence in Section 2.1:

"The overall aim was to assess realistic implementation of specific mitigation measures."

As well as the emissions methodology and data being published elsewhere it appears that the CMAQ modelling results have also essentially been published elsewhere. There is another recent paper by Pommier et al. with title "The Impact of Farming Mitigation Measures on Ammonia Concentrations and Nitrogen Deposition in the UK" published in Atmosphere 2025, 16(4), 353; https://doi.org/10.3390/atmos16040353. This other paper already describes (i) the development of the same potential mitigation measures as described in this paper, (ii) the results from the CMAQ modelling of these mitigation measures on particulate NH4+ and PM2.5 concentrations across the UK, and (iii) an explanation for the rather low reductions in the latter two entities simulated by the modelling. Whilst it could be argued that the current paper presents a little more data of the authors' CMAQ modelling of PM2.5, the descriptions of the model limitations and of the authors' conclusions on the impact of their NH3 mitigation scenarios on UK PM2.5 concentrations is essentially the same.

To justify a bit more this second modelling study, it is worth noting the previous study Pommier et al. (2025) only focused on regional modelling.

It is true that few outcomes are similar, i.e. the limited impact of the NH3 emissions scenarios on the secondary PM2.5 formation.

However, this 2nd study completes the 1st study published in Atmosphere journal, by providing details on the methodology applied to build the emission scenarios and using local modelling at farm levels in addition to the regional modelling

Notably, all this information could not be condensed into a single, easily digestible paper.

**For all these reasons, we have decided to split the study into 2 parts.**

In relation to the CMAQ modelling part of this work, Section 3.1.1 of the present paper presents rather poor model simulation of present day (2019) annual mean PM2.5 at rural PM2.5 monitoring sites. The model values underestimate measurements by a factor of two on average, and the spatial correlation across 48 measurement sites is only 0.58, which implies an explanation of variation of only 36%, i.e. well under half of the measured spatial variation is captured by the model. The authors do not describe further diagnostic model-measurement comparisons. I checked the previous Pommier et al. paper referred to above, and found that its Supplementary Material section does contain some other model-measurement comparisons, for NH4+ and for NH3; but the model-measurement comparisons for both these species essentially have no correlation at all. It is surely a concern for a study whose focus is on the link between NH3 emissions and the effect on NH4+ and PM2.5 that the model simulations are so poor. The author state that their model performance statistics "are not fully satisfactory" but then carry on with presentation of the results from this "not fully satisfactory" modelling. The reader requires far more convincing from the authors that data from their CMAQ modelling yields reliable insight.

We fully acknowledge that the simulation of annual mean PM2.5 concentrations at rural monitoring sites in 2019 shows notable underestimation and limited spatial correlation. These limitations are indeed important and we agree that they warrant careful consideration. Our statement that the model performance statistics are "not fully satisfactory" was intended to be transparent about these shortcomings. However, we appreciate that this may not sufficiently convey the implications for the reliability of the subsequent analyses.

Moreover, these limitations are also detailed in the discussions section and assumptions on the underestimation were given.

It is also worth reminding, as cited in the paper, 2 recent modelling papers (Marais et al. 2023 & Kelly et al. 2023) had to tweak the similar inventory (NAEI 2019) to obtain a reasonable PM2.5 modelling (r=0.66, NMB=-11%). However, we have chosen not to apply such adjustments strategy, as they still do not perfectly align with observations.

The one aspect of the current paper that doesn't appear to have been published elsewhere is the ADMS dispersion modelling of fall-off of NH3 and PM2.5 concentrations in a few km distance from individual farms. As one would expect, the concentrations fall-off rapidly, down to 10% of max concentration within 1000 meters or shorter. However, these results are caveated by the authors with a long list of sources of uncertainty or difficulty with the input emissions data (that are published elsewhere) so again I was left wondering about the reliability of these results. Regardless of the accuracy or otherwise of these dispersion modelling results I was left wondering what would catch the international reader's attention from the fact that emissions from a point source dilute quite rapidly. Instead, the main message appears to be that more data and research are needed.

It is correct that the measurements used for estimating the emissions used in the local modelling was not available when this manuscript was shared. However, the measurement paper is now available at: Leonard, A.; Wiltshire, J. Agricultural Emissions Measurements from Five English Farms. Preprints 2025, 2025082187. https://doi.org/10.20944/preprints202508.2187.v1

Yes, more data and research are needed. We have added this sentence in the conclusions:

"To evaluate the potential impact of these emissions on rural populations, one approach would be to map population distribution around agricultural holdings. This would help estimate the number of individuals likely to be exposed to such emissions."

The reviewer also needs to know that the measurements campaigns experienced major challenges. The study was conducted during a challenging period, beginning with the onset of the COVID-19 pandemic at the beginning of the project, which significantly affected the recruitment of farms for fieldwork. Engagement with pig farms was particularly difficult due to severe abattoir delays—partly linked to the UK's departure from the European Union—which led to overcrowding on farms. These factors caused substantial delays in recruitment, further compounded by the withdrawal of two participant farms that had to be replaced, and operational issues at another farm that prevented the collection of usable data. Additionally, concerns about infection risks—both from COVID-19 and general biosecurity—limited access to measurement equipment.

This information has been added in Section 2.3.2.

**Some specific comments:**

L46: Sentence starting "Resulting of" is not grammatically correct. It is also not clear to me what the point is that this sentence is making.

The sentence has been changed and now it reads:

"The UK presents a significant case for examining the influence of NH3 on PM2.5 levels due to its varied agricultural practices, transport-related emissions, and industrial activities."

L59: Sentence starting "Results confirmed" is not grammatically correct.

The sentence has been changed as:

"These results were confirmed by the study of Ge et al. (2023), since they showed NH3 reductions are more effective for regions or countries with better air quality,"

L63: The statement here that Ge et al. (2022) suggests NH3 reduction only has minor improvement on PM2.5 in the UK contradicts what the sentence starting in L59 says.

Yes, it is correct. It is the reason the sentence starts by the word "Conversely". Not all studies agree on the best methodology for reducing  $PM_{2.5}$  concentrations.

L123: There is reference here and in Appendix A to the investigation of 19 mitigation measures to reduce NH3 emissions, yet Table A1 lists 38 different measures that it is stated were used in each of the three scenarios. To confuse the reader further, Table B1 lists 20 mitigation measures. How many, and which, mitigation measures were actually incorporated in the model emission scenarios?

We apologize for this confusion, and we thank the reviewer for spotting this error. The correct number of mitigation measures is 20. It has been corrected in the text.

Moreover, the number of measures in Tables A1 and B1 are different as several of the measures in Table B1 and are included in several livestock sectors, and so appear multiple times in Table A1.

For example, "Amending livestock diet to match N content" (6th measure in Table B1) is included once in Table B1, but is listed in Table A1 three times as it was applied to pigs, poultry, and dairy sectors.

We have added the following sentence in Section 2.1:

"The number of measures listed in Tables A1 and B1 differ because each measure appears only once in Table B1, whereas Table A1 includes measures multiple times when they apply to more than one livestock sector."

**L162: What is the word "This" at the start of this sentence referring to?**

Yes, the 'this' refers to the 11 measures. We agree the sentence that starts at line 162 is not very clear because of this, it has been amended to:

"To determine how to reflect these eleven measures in the SMT (including what stage(s) in the agricultural system the measure is relevant to and if it is an 'Emission' or a 'Reduction' measure), as well as the emission reduction potential of these measures, COGAP, BAT, and expert knowledge was used."

L215, and Figure 2: The explanation of why CO emissions decrease in the agricultural NH3 emission mitigation scenarios is not at all clear. Surely the NH3 emission reductions measures have essentially no impact on CO emissions, or if they do have some impact – such as account being taken of how combustion emissions may be reduced through the process of enacting the NH3 mitigation measures – then surely there would be reductions in NOx emissions (and possibly also in primary PM emissions) alongside the reductions in CO emissions? If there is some form of bias in the modelling of CO emissions, as the text seems to imply, then why weren't the CO emissions for the mitigation scenarios set to be the same as for the base2030 run?

It is right that the NH3 emission reductions measures don't impact CO emissions. This reduction in CO in the 3 scenarios are resulting from the absence of a few sources of CO in the SMT which are included in the NAEI. However, CO chemistry does not impact NH3.

L231: Simply stating that "filling" was used where data capture was poor is not helpful: what quantitative approach was used for data filling? Also, start a new sentence at "Where".

This paragraph has been rewritten as follow:

"For the local modelling, meteorological datasets were procured from National Oceanic and Atmospheric Administration (NOAA) weather stations ranging from 6km to 25km from farms in this study. Where data capture was insufficient, gap-filling was performed to ensure coverage exceeded 85% for all parameters, including wind speed, wind direction, cloud cover, temperature, and precipitation. Data filling involved selecting the most representative NOAA station for each farm, and where gaps were present in its dataset, missing values were supplemented using data from the next most representative station. This approach ensured a completer and more reliable dataset for modelling."

L234: This sentence mentions "Each farm..." but nothing has been mentioned about individual farms so far in this paper. What farms? There are positions of some farms marked on a map in Fig 2b but this map is never referred to in the text.

We have changed the sentence (in bold) as below:

"Each farm, situated in a different region of the UK (Fig. 1), away from major roads and industrial areas, had a 15km-by-15km points grid centred at the farm with a 100m resolution."

We have also added this sentence in Section 3.2:

"These farms were in different parts of the UK as shown in Figure 1."

Please note, as stated in lines 256-257 that "These farms have remained anonymous for the study".

L275-L297: it is very difficult to follow what nature of temporality in emission profiles was actually developed and applied in the ADMS model for which particular sources on each farm.

This section has been updated to clarify the methodology.

For example, we have modified the following sentence: "An order of preference for time varying emission profile development has been implemented. The most preferred to least preferred was defined as below:"; in addition to the paragraph as explained in the 2 next comments.

L275-297: It appears that no seasonality in NH3 emissions were applied to most, or all, of the outdoor sources of NH3, yet surely outdoor NH3 emissions are very temperature, i.e. season, dependent.

For Farm 5, two emission rates were used to reflect seasonal housing and grazing periods, acknowledging the season shift in emissions due to extended grazing (see lines 289-294). However, for most other sources there was a lack of detailed measurement data so a constant emission rate was applied throughout the year, which may not fully capture seasonal variability. This decision to use constant emission rates was due to limited temporal coverage in the measurement campaign, which is noted as a limitation in the study (see lines 466-471).

L290: The sentence starting "Loafing areas" cannot be understood.

To reply to the previous comment and this comment, this part of the manuscript has been rephrased as below:

"On farm five, the milking and loafing area was the only building where emission measurements were taken and the only one for which a diurnal profile was applied. These loafing areas were non-passageway, non-feeding spaces where cows can lie down and move freely, allowing them to express natural behaviours such as grooming and heat detection. Grazing areas and cattle housing were assigned two distinct emission rates to reflect seasonal differences between periods when cattle are grazing and when they are housed."

L315: Presumably the authors mean that the value 0.58 they quote here is the ratio between PM2.5 and PM10 concentrations, but they write it the other way around.

Agreed and we thank the reviewer for seeing this error. We have corrected the sentence, and it is now "the ratio between  $PM_{2.5}$  and  $PM_{10}$ ".

L340-L344: The fact that increasing NH3 emissions by 50% doesn't increase PM2.5 doesn't automatically imply that present-day NH3 emissions are not limiting to PM2.5. To address that question requires simulating with lower NH3 emissions than present-day NH3 emissions.

It is a good point. The test on the 50% increase of NH3 emission follows the suggestion in the correction of NH3 UK emissions given by Kelly et al. (2023) and Marais et al. (2023).

The lack of a PM2.5 response to a 50% increase in NH3 emissions indeed doesn't rule out the possibility that current NH3 levels are still limiting PM2.5 formation under real-world conditions. As you suggest, the key test would be to simulate scenarios with reduced NH3 emissions—ideally below present-day levels—to observe whether PM2.5 concentrations decline.

However, one of the main novel aspects of the study is on developing realistic scenarios based on input from a range of stakeholders etc. In other words, rather than looking at a certain percentage reduction in NH3, these estimates are grounded in specialist input from farmers, academia, and advisers.

Thus, despite unrealistic large reductions (e.g. a global reduction of 50% in UK emissions), the focus of the study was to assess the impact of realistic implementation of measures (even with small total emission reduction) on pollution.

L357: I'm sure the authors quote a MRE value here that is too small by a factor of 100. As their NMB value is -51%, I presume there MRE value should read  $\sim$ -50% here, rather than  $\sim$ -0.5%. It would not be possible to have NMB and MRE values for the same dataset that differ by a factor 100.

Thank you for having seen this mistake. The MRE is in unitless (ratio); or should be multiply by 100% to give a % value.

This has been corrected.

L563: There is an error here, the mean relative error does not have unit of concentration. If expressed as a ratio then it is unitless.

Yes, it has been corrected, as for the previous comment.